# Drug Delivery Nano-Platforms for Advanced Cancer Therapy

Ekaterina Naumenko *, Ivan Guryanov and Marina Gomzikova

Institute of Fundamental Medicine and Biology, Kazan Federal University, 420008 Kazan, Russia;
ivan.guryanov@gmail.com (I.G.); marina.gomzikova.gmo@gmail.com (M.G.)
* Correspondence: ekaterina.naumenko@gmail.com

**Abstract:** The incidence of cancer is growing every year and covers all age groups, including the working population, which makes cancer socially significant. Existing methods of treatment, despite the effectiveness of individual compounds in relation to cancer cells, are not perfect due to a number of side effects associated with high doses that physicians are forced to administer when using treatment protocols. A particularly difficult issue is the creation of effective functional containers that would have the properties of targeting certain types of cells. The solution of this problem is currently relevant, which is reflected in the growth of publications on this subject in recent years. The most promising is the use of nanotechnology in the development of bioengineered therapeutics and containers for chemotherapeutic agents. In this review, we tried to assess the trends that exist in this area of research, as well as show the wide using of some commercially available formulations based on the nano-sized vehicles.

**Keywords:** cancer; drug delivery systems; metal nanoparticles; nanotubes; liposomes; dendrimers

## 1. Introduction

Cancer is a socially significant disease that affects various populations around the world. Of primary importance is the impact of oncological disorders on the disability of the able-bodied part of the population and overall mortality. Cancer may be defined as multifaceted disease affecting various organ systems. According to the International Agency for Research on Cancer (IARC) cancer is a leading cause of death worldwide with 10 million deaths in 2020, corresponds to nearly one in six deaths. At the same time, there is an increase in the number of patients, which, among other factors, is influenced by improved diagnostics. Approximately 19 million cancer cases detected in 2020, with new cases predicted to double by 2040 [1].

Specific cancer diagnostics and effective treatment are becoming important challenge for medicine and pharmacological science and industry in recent decades. When developing new forms of drugs, it is important to take into account the targeting of their delivery and release controlled by various factors, reducing the effect on healthy cells and systemic side effects. The founder of targeted delivery is considered to be Paul Ehrlich, one of the stages of whose scientific career was associated with the search for chemotherapeutic agents for the treatment of infectious diseases (such as syphilis) and cancer [2]. At the beginning of the twentieth century (1910), he proposed the concept of the "magic bullet", which was that one part of the drug should recognize and bind to a target in the body, while the other part should exhibit therapeutic activity against that target.

Recently, the search and synthesis of drugs has undergone great changes due to the accumulation of a massive pool of data on the action of various compounds and the creation of computer programs aimed at discovering new drug formulations with active centres and ligands to increase targeting and reduce adverse side effects. However, despite all these developments, anticancer therapeutics still have a strong effect on surrounding tissues and damage them, in addition, the formation of multi-drug resistance (MDR) in tumor cells is an important aspect. MDR is represent the ability of cancer cells to survive after

treatment using different types of anticancer drugs [3], match to the concept commonly applied to antibiotic therapy [4]. Despite the effectiveness of a number of anticancer drugs in the initial treatment, many cancer patients may develop resistance during continued treatment [5,6]. Studies of the development of this kind of resistance of cancer cells to treatment have shown that cancers cases already in remission often tend to relapse. For example, 30–55% of patients with non-small cell lung cancer (NSCLC) relapse [7], while for ovarian adenocarcinomas recurrence is observed in almost 25% of cases with early-stage diseases and in more than 80% with more advances stages [8,9]. In children with acute lymphoblastic leukemia, recurence reaches up to 20% of cases after the development of stable remission [10]. Thus, the need for innovation to improve the functionality of existing drugs through delivery systems becomes apparent. The development of new drug forms based on nanoparticles, liposomes, stem cells and vesicles, erythrocytes and other similar carriers is aimed to reduce all negative phenomena connected with repeated malignant transformation of cells, treatment process and targeting.

There are some approaches that can help create controlled release systems for effectively and safely treating diseases. One of the tools for achieving targeted and controlled medicine release is the use of controlled release formulations. These formulations can be designed to release the medication at a specific place, ensuring that the drug remains at therapeutic levels in the body for an extended period of time. It can be achieved simply by specific materials used for drug encapsulation [11]. Smart materials are a cutting-edge tool that can be used to achieve targeted and controlled medicine release. These materials can respond to changes in the body, such as pH or temperature, and release the medication accordingly. This allows for precise control over the release of the medication and can improve the effectiveness of the treatment [12]. Hydrogels are three-dimensional networks of crosslinked polymer chains that can absorb and retain large amounts of water. These materials based on natural polymers like gelatin, dextral, glucans as well as synthetic like N-isopropylacrylamide, acrylamide, polyvinyl alcohol and polyethylene glycol have found applications in drug delivery due to their ability to release medication in a controlled manner [13]. Hydrogels can be designed with sensitivity to external stimuli (temperature, magnetic field, pH etc.), loaded with drugs and implanted at the target site, providing sustained release over an extended period.

Controlled drug release, on the other hand, involves delivering the medication at a controlled rate over a period of time. Bioelectronic Micro-System, the innovative platform of controlled drug delivery, can solve this task effectively. This cutting-edge technology combines the principles of biology and electronics to create highly efficient and targeted drug delivery systems that can have a profound impact on patient care and outcomes [14].

One of the most effective tools for achieving targeted and controlled drug release seems to be the use of drug delivery based on nanosized materials. The special properties of these materials make it possible to create dosage forms with a slow release of the drug over time, providing targeted delivery to specific cells or tissues and enhanced penetration ability (Figure 1).

Highlights of nano-sized drug delivery systems in comparison with conventional systems include:

- Enhanced targeting: nanoparticles can be engineered to target specific cells or tissues, allowing for precise drug delivery.
- Improved drug solubility: many drugs have poor solubility, which can limit their effectiveness. Nanoparticles can improve the solubility of these drugs, ensuring better bioavailability.
- Extended drug release: nanoparticles can be designed to release drugs slowly over time, leading to sustained drug levels in the body.

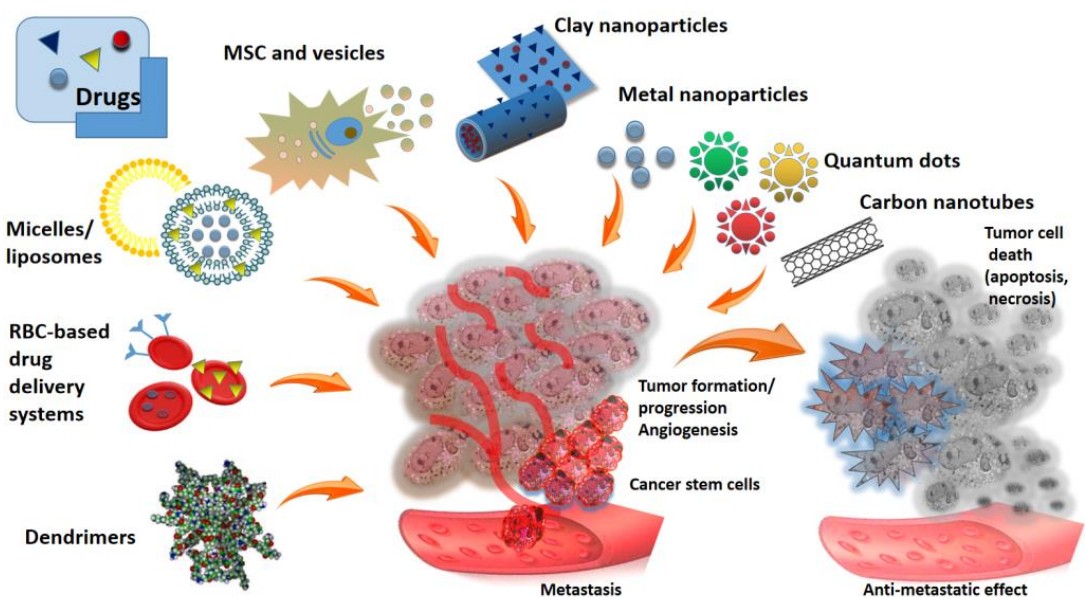

**Figure 1.** Schematic overview of different anticancer drug delivery platforms and tumor organisation.

While nanoparticle medicine delivery systems offer numerous advantages, conventional drug delivery systems still have their place in the medical field. For example, oral tablets and capsules are convenient and easy to administer, making them suitable for many patients. Additionally, some drugs may not benefit from nanoparticle delivery systems, making conventional methods a better choice.

Nanoparticles as a carrier may have a synergistic effect to enhance anti-cancer properties of drugs. The biological and physicochemical properties of anticancer substances that are supposed to be used in combination with nanomaterials determine the design and engineering of carriers for them. Thus, the physicochemical and functional properties of nanosized carriers vary for each antitumor agent [15]. A broad selection of nano-constructions and a wide range of their combinations with auxiliary materials allows more accurate distribution of drugs in tissues affected by tumors and fine control of the mechanism of therapeutic action [16].

The preparation of nanomaterials requires precision and expertise to ensure optimal drug delivery outcomes [17].

Several techniques are utilized to fabricate nanomaterials, including:

- Bottom-up approach [18]. This method involves building nanoscale structures from atomic or molecular components. It allows for precise control over the size and composition of the nanomaterial.
- Top-down approach: in contrast, the top-down approach involves reducing bulk materials to nanoscale dimensions. The top-down method includes physical participation approaches such as mechanical processing, physical vapor deposition (PVD), lithography and pyrolysis [19]. This technique often results in uniform nanomaterials with consistent properties.
- Self-assembly: nanomaterials can also be formed through self-assembly processes, where molecules spontaneously organize into ordered structures. This technique is advantageous for creating complex nanoscale architectures. Spontaneous association of individual blocks through self-assembly can lead to the formation of ordered structures ranging from angstroms to centimeters of various sizes and shapes. Self-assembly of amphiphilic nanostructures such as micelles, vesicles and hydrogels occurs through various physical interactions. Self-assembled nanostructures have shown great potential to be used as simple and effective materials for this purpose [20].

Nanoscale drug delivery systems are usually based on the incorporation (encapsulation or surface immobilization) of anticancer compounds into biocompatible carriers.

Nanomaterials in such composite preparations ensure the stability and efficiency of bioactive molecules [21]. The tasks of targeted distribution and penetration of drugs are solved due to the submicron size of carriers, the possibility of modifying the surface of nanomaterials with compounds with different adhesiveness to the cell membrane (polyelectrolytes, specific antibodies and ligands, etc.), which makes it possible to model the targeted interaction, as well as controlled release of drugs. Aptamers can also be used as smart ligands for targeted drug delivery [22]. Composite preparations containing nanomaterials are already used in clinical trials and are gradually entering the pharmaceutical industry market [23].

Another problem of targeted delivery of chemotherapeutic drugs is the controlled release of the loaded substance at a given locus. To prevent premature evacuation of drugs from the lumen of nanotubes, functional plugs (stoppers) are formed at the ends of tubular containers [24]. Along with controlled release, the accuracy of the interaction of carrier nanoparticles with tumor cells remains an important task. As a rule, targeted delivery is provided by compounds mobilized on the surface of nanoparticles. Such interaction mediators or vector molecules can be folic and lactobionic acids [25–27], peptides (Arg-Gly-Asp) [28], and antibodies [29]. The direction of the interaction can be provided by the pathophysiological features of tissues affected by tumors. such as enhanced permeability and retention effect (EPR), which leads to the accumulation of large molecules and small particles. For example, halloysite nanotubes (HNTs)can also accumulate at the tumor site based on this effect [24].

Nanomedicine is the relatively new but extensively developed branch of medicine that utilizes the nanotechnologies for treatment of various diseases using the nanoscale objects, such as nanoparticles [30] and nanorobots [31]. Nanotechnology overcomes drug delivery barriers such as low drug solubility, non-specific bio-distribution, low bio-availability, tissue barriers, side effects, and lack of specificity. In turn, cancer nano-therapeutics provide high sensitivity, specificity, and multiplexed measurement capacity [32]. All this contributes to the creation of convenient dosage forms, a reduction in the concentration of the active substance, which in turn reduces toxic effects during treatment, prolongs the drug life cycle. Various types of drug nanoformulations have been created by researchers and are already undergoing clinical trials [33], however, each specific case of a disease and a specific drug requires its own delivery systems, their modifications for better targeting certain cancer cells, and therefore this area of science is actively developing despite specific successes.

Nanoparticles can penetrate into the tumor through passive and active processes (Figure 2). The first is due to the fact that in the tumor there is a network of abnormal branches of blood vessels with leaky areas with a pore size of 100 nm, which facilitates their penetration. This occurs due to uneven growth of vascular wall cells: there is a rapid growth of endothelial cells and a simultaneous decrease in the number of pericytes [34]. In this case, passive targeting is observed, called the effect of increased permeability and retention. However, the active mechanism is most preferable, since it allows the selective accumulation of nanoparticles in the tumor by modifying their surface with tumor-targeting ligands (for example, folic acid, peptides, antibodies, etc.) [35].

Design of nanoscale drug delivery systems represents the most perspective technology in the area of nanoparticle applications due to their possibility to modify properties like solubility, drug release profiles, diffusivity, bioavailability and immunogenicity [36]. Choice of ideal nano-drug delivery system primarily must be based on the biophysical and biochemical properties of the drugs being selected for the treatment [37,38]. In addition to biocompatibility and other aspects mentioned above, nanoparticles used to create nanodrugs and as additive systems must meet size requirements. In particular, systemically distributed nanoparticles should be greater than 10 nm to prevent premature excretion by kidneys. In addition, most of researchers suggest it is important that they should be also less than 200 nm to be able to pass through the microcapillaries without producing embolism [39,40]. An important parameter is also the polydispersity index (PDI), the value of which should be about 0, which indicates the monodispersity of the sample, while the

slightest variations in PDI can lead to serious increases in toxicity and decreases in the biocompatibility of nanomaterials [41].

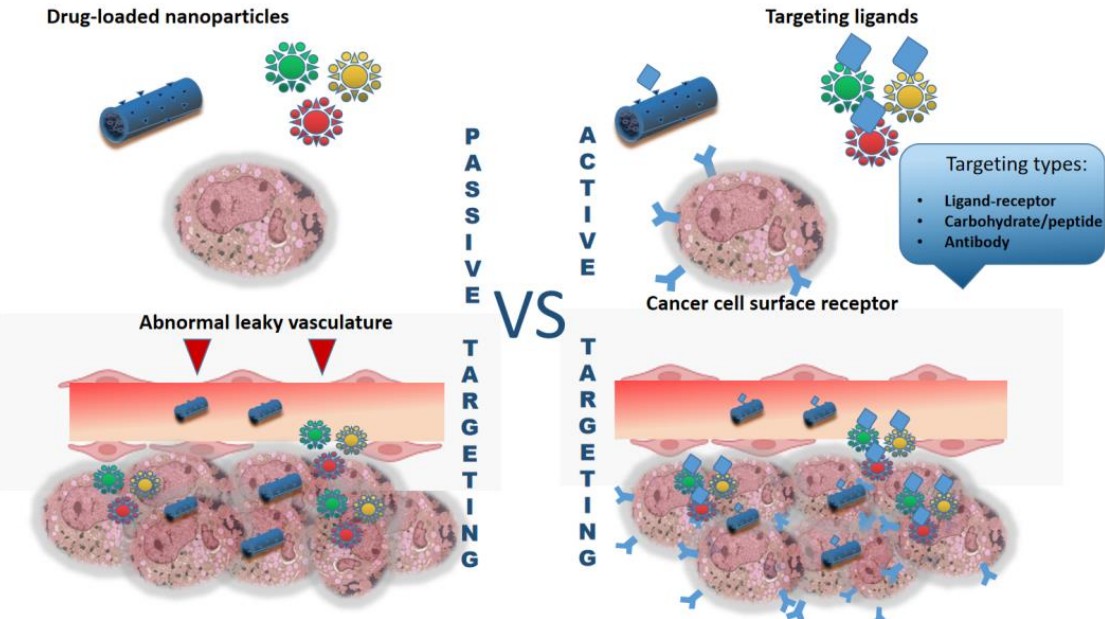

**Figure 2.** Brief scheme of active and passive targeting of nano-formulated grug-delivery systems.

The fabrication of nanosized carriers and drug formulations has the main goal of reducing side effects, so it is important to take into account the intrinsic toxic effect of nanoparticles. Recently, such a direction as the green synthesis of nanoparticles has been actively developing, which, as shown by a number of studies, can reduce the toxicity of the nanoparticles themselves and optimize the process of their formation from the point of view of the environmental friendliness of the process [41–44].

In this review we summarized the latest trends, some clinical trials and perspectives innovative of drug delivery systems based on biocompatible nanosized to improve the targeting of cancer therapy. The purpose of this paper is to create a concise digest of currently existing researches and finished products in the field of drug delivery for scientists, especially those involved in the development of nanosystems in terms of biomedical applications. A particular emphasis will be given to the clinical use of certain drug delivery systems, their advantages and drawbacks.

## 2. Metal Nanoparticles as a Vehicles for Anticancer Therapeutics

Metal nanoparticles in a number of studies are recognized as an effective tool for cancer treatment not only as drug delivery agents but also as boosted agents for the phototermal therapy and in radiation therapy. Nanoparticles of both noble and non-noble metals such as gold, silver, iron, copper etc. and their oxides are already actively used in clinical trials and in recent years there has been an increase in publications devoted to these studies [45]. However, there are a number of limitations imposed by their use in the organism. For example, it is important to take into account the period of removal of nanoparticles from the body, their distribution, the level of accumulation in target and non-target areas, the absence of toxic effects, as well as reducing the environmental impact by reducing the use of harmful solvents and switching to green synthesis. Here we focus on just two, but most popular metal nanoparticles for cancer therapy: iron and gold.

### 2.1. Iron Oxide Nanoparticles

The tumor can be directly exposed to near infra-red light or oscillating magnetic fields. In this case, the action of the magnetic field can be directed to the formation of reactive oxygen species or its influence can be mediated by magnetic iron oxide nanoparticles, which leads to the formation of hyperthermia in the tumor area with its subsequent destruction [46]. The use of superparamagnetic nanoparticles makes it easy to control them as using locally acting external magnetic field as due to covalent binding with molecular determinants characteristic of a certain type of tumor.

Magnetic nanoparticles (MNPs) can be used for nano-labelling of living cells and their safety was shown in experiments in vitro and in vivo on different types of cells and organisms [47] (Figure 3). It has been shown, in general, that positively charged MNPs interact strongly with blood components and are cleared relatively quickly from the systemic circulation, contrasting to negatively and neutrally charged MNPs. Wherein the most optimal size of MNPs for in vivo use is within the range of 15–100 nm in diameter [48].

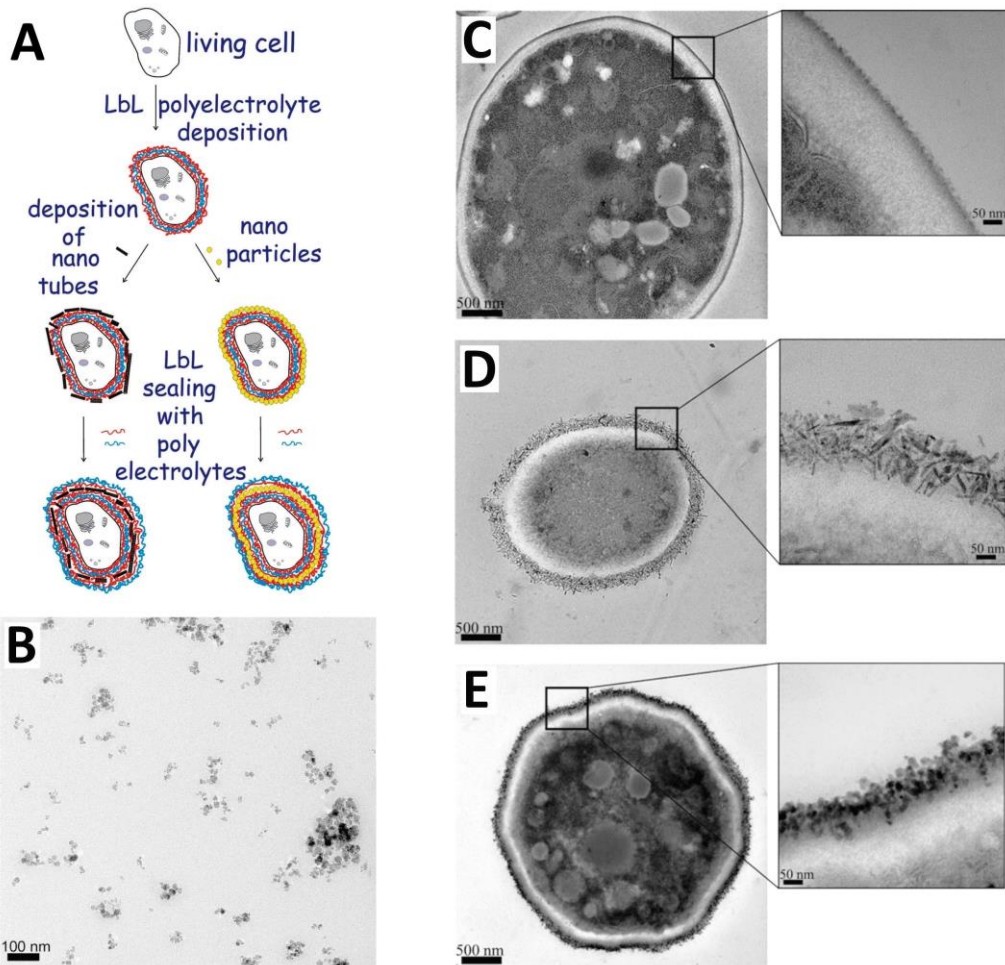

**Figure 3.** Scheme of polyelectrolytes-modified Layer-by-Layer deposition of iron nanoparticles onto the cell surface (**A**) (reproduced with permission from [49]; typical TEM image of polyelectrolyte-stabilized magnetic nanoparticles (**B**); reproduced with permission from [50]; transmission electron microscopy images of thin sections of (**C**) native yeast cells; (**D**) yeast cells coated with polyelectrolytes-stabilized magnetic nanorods and LbL coated with polyelectrolytes layers; and (**E**) yeast cells coated with series of polyelectrolytes/polyelectrolyte-stabilized magnetic nanospheres. Reproduced with permission from [51].

An important aspect is the ability of MNPs to accumulate in certain areas and contribute to the directed formation of hyperthermia, which eliminates the impact on healthy tissues. To increase the functionality, iron oxide nanoparticles can be modified with various coatings, which makes it possible to increase their stability, reduce the rate of loss of magnetic properties, and increase affinity for certain cells [52].

Iron oxide nanoparticles have a fairly long history of study as therapeutic and diagnostic agents, and their activity in relation to the immune system has been revealed due to interaction with immune cells and activation of various factors [53]. Nanoparticles for imaging and MRI applications, for example, superparamagnetic iron oxide nanoparticles (SPIONs), must avoid immune recognition and increasing of inflammatory factors secretion, oppositely for the cancer treatment applications and therapeutic nanoparticles fabrication controlled immune response will be useful feature. At the same time, the ease of surface modification of nanoparticles makes it possible to create various functional coatings, which, among other things, affect their immunogenic properties and toxicity as well as colloidal stability. Lazaro-Carillo et al. demonstrated the biocompatibility of tailor-made PEG-coated iron oxide nanoparticles (PEG-IONPs) in vitro and in vivo and tested their properties as MRI contrast agent [54]. At the same time, in the work of Escamilla-Rivera et al. PEG-IONPs increased the level of reactive oxygen species (ROS), tumor necrosis factor alpha (TNF$\alpha$), interleukin 1 beta (IL1$\beta$), and mitochondrial dysfunction in THP-1 human monocytes [55]. Starch-coating of IONPs as reported Gonnissen et al. can alter subtle features, such as the cytoskeleton and ion channel functions, in monocytes without any change in IL1$\beta$ and IL10 secretion [56]. Polyvinyl alcohol (PVA)–coated IONPs induced IL1$\beta$ levels in monocytes without affecting their survival [57]. Karimova et al. demonstrated that chitosan and polyethylene glycol coatings affect the structural properties, drug-loading efficiency, and anti-cancer efficacy of $Fe_3O_4$ nanoparticles [58]. In the study of Bhattacharya et al. design of dual temperature and pH responsive polymer integrated magnetic nanohybrids comprising of smart block copolymers and mixed ferrite nanoparticles for efficient anti-cancer drug delivery and magnetic resonance imaging [59].

At the same time dextran-coated ultrasmall SPIONs (30 nm) did not trigger inflammatory responses on primary human macrophages and have no impact on secretion of IL-12, IL-6, TNF-a and IFN-c by the cells [60]. Release of TNF-a in primary human macrophages under exposure of dextran-coated SPIONs and silica-coated SPIONs also not observed in the work of [61].

Currently, spherical iron oxide nanoparticles are used in clinical practice for hyperthermic therapy of glioma and glioblastoma and prostate cancer [62–64]. U.S. Food and Drug Administration (FDA) approved for the therapy and diagnostics such iron oxide nanoparticles-based agents as Feraheme® for iron deficiency treatment; Combidex® (U.S.) and Sinerem® (Europe) as a magnetic resonance imaging (MRI) agent; Nanotherm® (MagForce) for cancer treatment; and Lumirem® as an oral gastrointestinal tract imaging agent [65].

According to a number of studies, Feraheme® (ferumoxytol) has a wide potential and can be used for cancer therapy and diagnostic procedures. Thus, Trujillo-Alonso et al. demonstrated that ferumoxytol treatment results in a significant reduction of disease burden in a murine leukaemia model and patient-derived xenotransplants [66].

For cancer therapy, a number of drugs are used that can be conjugated with magnetic nanoparticles to increase the effectiveness of the drug and create a protocol for its targeted delivery directly to the tumor, for example, by generating a magnetic field. For example, Doxorubicin, as an FDA-approved agent, is by far the most frequent anticancer drug loaded onto magnetic NPs. The effect of doxorubicin-loaded magnetic NPs on various tumor types has been studied: breast [67], lung [68], and colon [69]. At the same time, nanopreparations were synthesized and functionalized by many different methods. In addition, other anticancer drugs such as paclitaxel [70], methotrexate [71], and epirubicin [72] also explored for loading and conjugation with magnetic nanoparticles for cancer treatment. Despite the progress made, there are a number of difficulties to

be overcome, such as the development of side effects in healthy tissues, as well as the development of drug resistance.

*2.2. Gold Nanoparticles*

Gold nanoparticles (AuNPs) are microscopic clusters of gold atoms that measure between 1 and 100 nm in size. Their small size and specific surface chemistry allow them to interact with biological systems in remarkable ways. Their size, shape, and surface properties can be tuned to design nanoparticles with specific functionalities for various applications.

AuNPs have garnered extensive attention in cancer therapy due beneficial physico-chemical properties, customizable size and shape. Geometrical parameters and catalytic abilities of AuNPs can be controlled through simple synthetic approaches. AuNPs enhance the accuracy of diagnostic imaging techniques, enabling early detection and precise monitoring of cancer progression [73]. Large surface area and high surface activity of AuNPs endow the ability of functionalization and variety conjugating with anticancer drugs and ligands. Nowadays, AuNPs have been exploited many therapeutic and diagnostic applications, particularly, as drugs and nucleic acid delivery, photodynamic therapy, photothermal therapy for tumors suppression and X-ray-based computed tomography imaging. The use of AuNPs as delivery tools successfully applied for delivery multiple types of antitumor molecules, including synthetic compounds [74], phytochemicals [75,76], therapeutic peptides [77,78].

Paciotti and co-workers in 2004 reported for the first time use of colloidal gold as delivery nanovectors [79]. Conjugated with macrolides antibiotics (azithromycin, clarithromycin, and tricyclic ketolide) gold nanoparticles (AuNPs) can accumulate in tumor-specific macrophages and induce their cytotoxicity, causing tumor cells death [80]. Also, AuNPs have been conjugated to a variety of antitumor substances, including paclitaxel [79], methotrexate [81], daunorubicin [82], gemcitabine [83], 6-mercaptopurine [84], dodecyl-cysteine [85], platinum complexes [86], doxorubicin [87], camptothecin [88], curcumin [89] and many other anti-cancer ligands [90] (Figure 4).

AuNPs conjugated with therapeutic nucleic acid (AuNPs-NA) can be used for gene suppression of expression of HER2 and ER$\alpha$ gene in breast cancer cells [91]. The complex AuNPs-NA has the ability to become a dual antitumor action nanoplatform that achieves simultaneously gene silencing and photothermal therapy [92]. Nanoparticle-mediated photothermal therapy, which NP employs as photothermal conversion agents to absorb near infrared (NIR) light and converts it into heat to ablate cancer cells, has been widely studied for cancer treatment. Bovine serum albumin (BSA)-coated gold nanorods (BSA-coated AuNRs) showed high photothermal conversion efficiency and a good photothermal ablation effect toward breast tumor cells [93].

Synthesis of gold nanoparticles (AuNPs) in phytochemical solutions has become tremendously prominent in biomedical applications. AuNPs synthesized in plant water extracts demonstrated conferred selective cytotoxicity against colon (Caco-2), breast (MCF7), and prostate (PC3) cancer cells and did not display any cytotoxicity to skin fibroblast and human embryonic kidney (HEK293) cells [94].

Thus, the chemical inertness and unique physicochemical characteristics of gold makes it a promising material for various biomedical applications including cancer treatment and therapy. AuNPs are distinguished by the ease of surface modification, the formation of complexes with pharmaceuticals, and the ability to form functional clusters. The potential of gold nanoparticles in cancer therapy has gained significant attention, leading to some clinical trials exploring their effectiveness, including the treatment of salivary gland tumors, advanced solid tumors therapy of primary or metastatic lung tumors etc. [73].

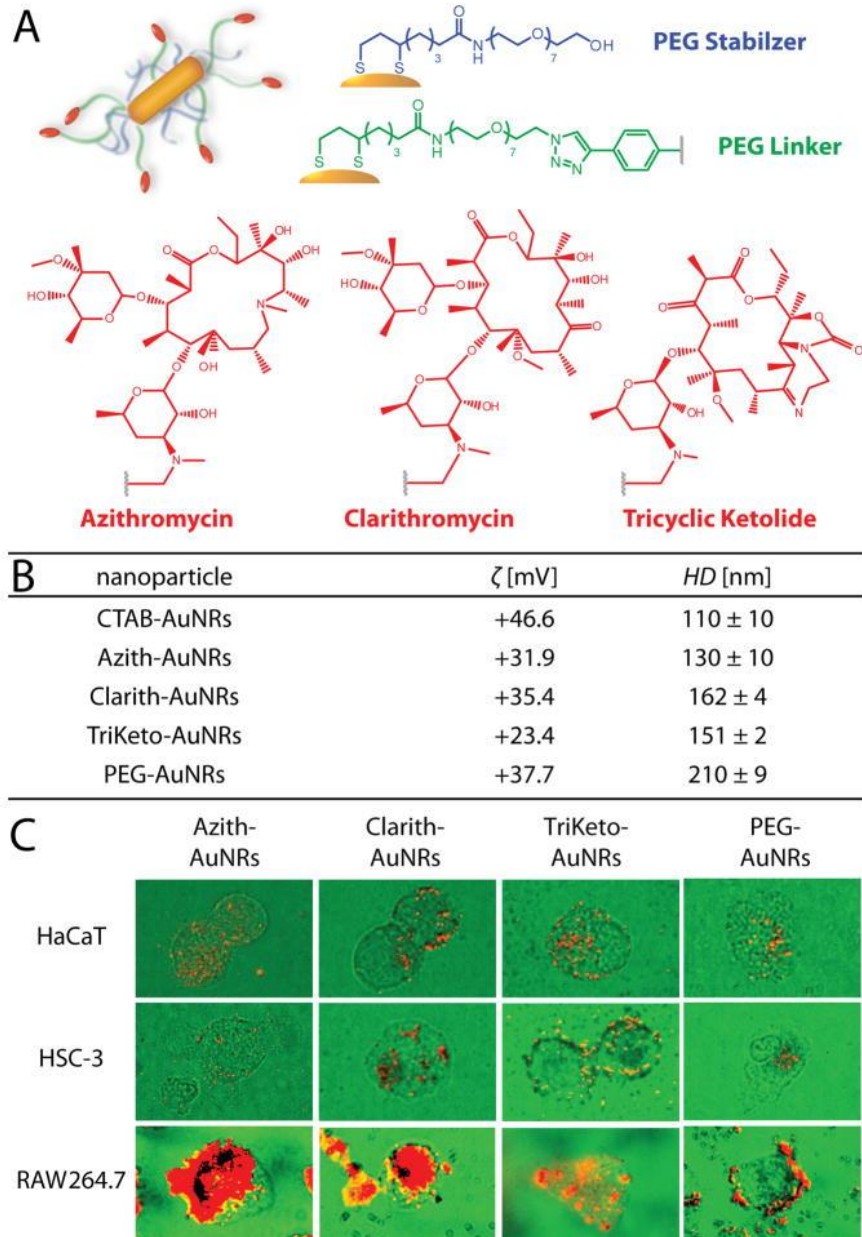

**Figure 4.** Schematic representation (**A**,**B**) physiochemical characteristics of the macrolide-gold nanorods used herein. Cardioid immersion dark-field scattering microscopy (DFSM) of cell cultures (green) illustrating preferential uptake/accumulation of macrolide-gold nanorods (red) into tumor-associated macrophage cells (RAW 264.7) relative to squamous cell carcinoma (HSC-3) and keratinocyte cells (HaCaT) (**C**). Reproduced with permission from [80].

### 3. Nanotubes as Drug Carriers

Nanotubes of different nature represent very promising drug delivery platform due to their shape. Their tubular structure makes it possible to load the internal cavity with various macromolecules, such as drugs, proteins, nucleic acids. In addition, due to the large surface area, therapeutic and diagnostic agents can be not only loaded inside the cavity, but also anchored on the outer surface. Loaded compounds can be released into the delivery area and this process can be preciously tuned by modifications of nanotubes' ends and surface by various molecules, including enzyme-activated stoppers and wrapping, as well as specific compounds pholate of hialuronic acid to enhance targeting of cancer cells [95].

Since 2000 and to the present time, researchers have remained interested in nanotubes of various natures, as shown by an analysis of publications on this topic in the PubMed system. Over twenty years, the number of publications has grown almost 40 times with a peak in 2021. It should be noted that this term includes carbon nanotubes, both single- and multi-walled, as well as mineral nanotubes and supramolecular artificial nanotubes assembled on the basis self-assembly (Figure 5) [96]. All these structures can potentially be used as systems for targeted delivery of anticancer drugs. We will consider below the two types of nanotubes that are most actively studied from the point of view of the formation of drug containers: carbon and natural mineral halloysite nanotubes.

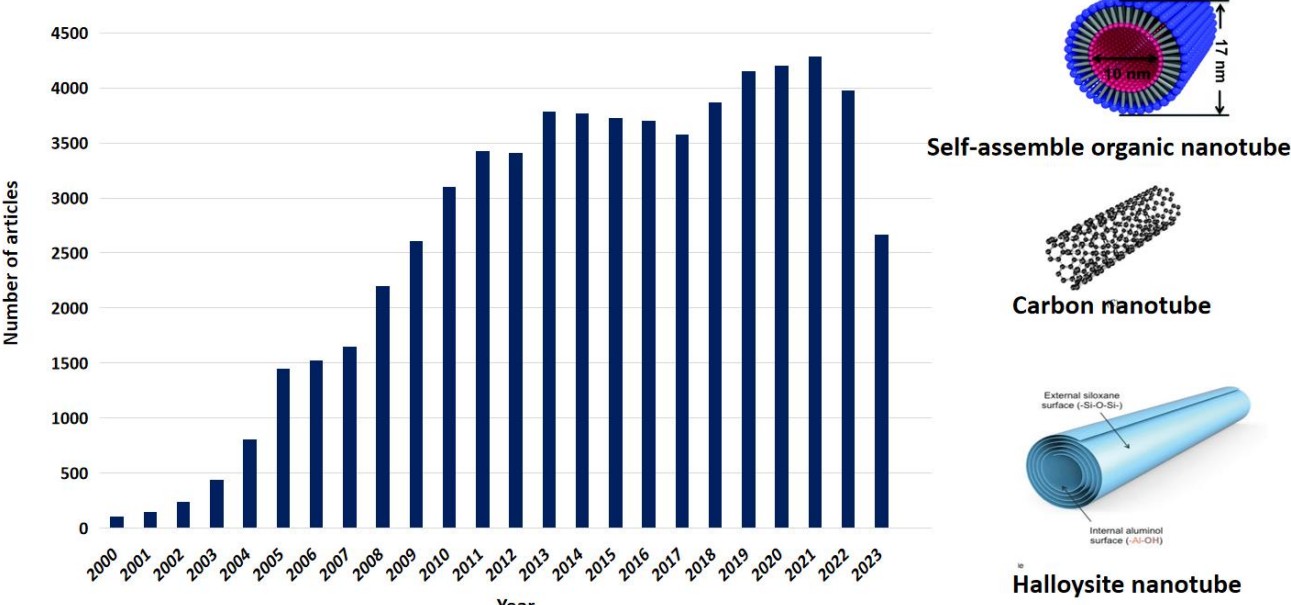

**Figure 5.** Publication activity related to the fabrication and use of nanotubes. In scheme illustration from [97,98] used under an open access Creative Commons CC BY 4.0 license.

### 3.1. Carbon Nanotubes

Carbon nanotubes (CNTs) are large cylindrical molecules consisting of a repeating pattern of sp2-hybridised carbon atoms, which may be formed by wrapping of single sheet of graphene into a cylinder called single-walled carbon nanotubes (SWCNTs). If multiple sheets of graphene rolling up, such construct named multi-walled carbon nanotubes (MWCNTs) [99].

Carbon nanotubes represent the promising drug delivery platforms that can be functionalized with a variety of biomolecules. This property allows for specific, targeted delivery of drugs to particular tissues, organs, or cells. Carbon nanotubes offer several advantages over other drug delivery systems due to their ability easily penetrate cells, delivering drugs directly to the cytoplasm or nucleus (Figure 6).

Different clinically approved anticancer drugs have been used to modify carbon nanotubes, and their effectiveness has been studied in vitro and in vivo systems (Table 1). The Table also shows the benefits that the drug acquires after loading into carbon nanotubes.

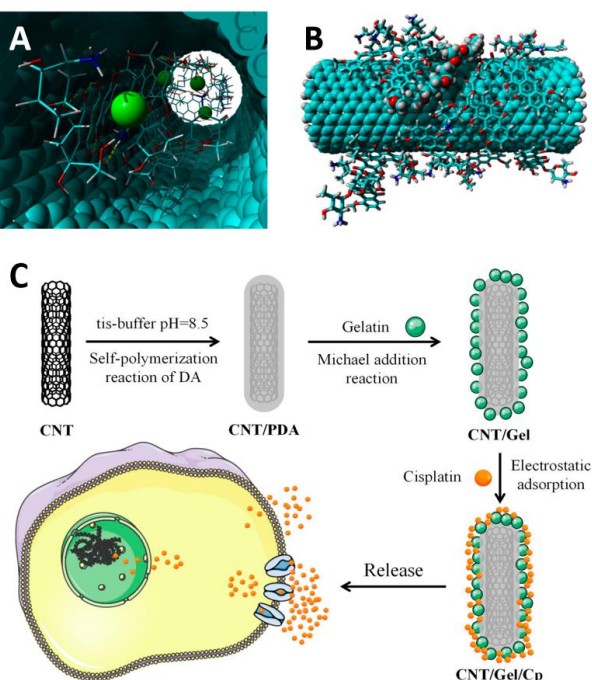

**Figure 6.** Localization of doxorubicin molecules inside CNTs (**A**) and on the outer surfase of nanotube (**B**) From [100] under an open access Creative Commons CC BY 4.0 license; (**C**) illustration of the process for the preparation of CNTs functionalized with cysplatin and its mechanism of action for antitumor effects. From [88] under an open access Creative Commons CC BY 4.0 license.

**Table 1.** Some anticancer therapeutics based on carbon nanotubes and anticancer drugs.

| Drug, References | Functionalization | SWCNTs/MWCNTs | In Vivo/In Vitro | Advantages |
|---|---|---|---|---|
| Cisplatin [101] | Gelatin | SWCNTs | In vitro | Precise and slow drug release |
| Doxorubicine [100] | Pyrrole Polypropylene Glycol | MWCNTs | In vitro | Better divisibility |
| Doxorubicine [102] | Hyaluronate | MWCNTs | In vivo | Higher tumor-growth inhibitory effect; absence of cardiotoxity, hepatotoxicity, or nephrotoxicity |
| Paclitaxel [103] | Chitosan Hyaluronan | SWCNTs | In vitro | Lower toxicity toward normal cells |
| Camptothecin [104] | Acid oxidation | SWCNTs | Not tested | Overcoming the insolubility and potential improving the efficacy while decreasing the adverse side effects |
| Methotrexate [105] | Carboxylation Polyethylenimine Folic acid | MWCNT | In vitro | Exclusive adsorbtion by cancer cells |
| Carboplatin [106]. | Carboxylation Folic acid | SWCNTs | Not tested | Improvemnet of folate receptor targeting |
| Carboplatin [107] | Not functionalized | MWCNT | In vitro | Enhancement of the toxic effects |
| Mitoxantrone [108] | Oxidation | MWCNT | In vitro | better delivery of the drug inside the cells |
| Ixazomib [109] | Polyethylene glycol | MWCNT | In vitro | Decreasing the toxicity of Ixazomib |

Depending on the disease, its stage and the chemical structure of the loaded drug, it is necessary to choose the correct method of delivering nanocontainers to the body. Carbon nanotubes can delivered to targeted locus by different routes, including oral, subcutaneous, abdominal, intravenous and intraperitoneal. Shorter carbon nanotubes are absorbed into the body more easily by oral route, passing through columnar cells of the intestines [110]. When delivered subcutaneously, the carbon nanotubes concentrate close to the injection site before slowly diffusing and being delivered through the lymphatic system, which may be useful for targeting metastatic cancer cells that migrate through this route. Meng et al. demonstrated

the significant activation of the complement system after subcutaneius distribution of water-soluble multi-walled carbon nanotubes, promotion of inflammatory cytokines' production and stimulation of macrophages' phagocytosis and activation [111]. The authors note that these changes in the immune system themselves may cause inhibition of tumor growth.

Intravenous administration of nanotubes leads to rapid distribution in the blood and delivery to various internal organs. It is important to consider that the size and chemistry of the surface, as well as various modifications of carbon nanotubes and their loading with drugs lead to changes in the retention time before elimination via the kidneys and liver. For example, functionalization by polyethylene glycol can improve the retention time of carbon nanotubes. Moreover PEG-modified CNTs have favorable pharmacokinetic and toxicology profiles and have been successfully tested in preclinical studies in the fields of oncology, neurology, vaccination, and imaging, for the fabrication of novel multifunctional nanodrugs [112]. PEGylated MWCNTs were created and used as a carrier for targeting the antineoplastic drug Ixazomib to myeloma cancer cells [95]. In this study the decreasing of the Ixazomib toxicity after loading to the MWCNTs-PEG composite was demonstrated.

An intraperitoneal method of introducing nanotubes has also been described, and although intraperitoneal injection is easier and faster to perform than an intravenous injection, it may result in unnoticed erroneous injections into the bowel or retroperitoneum as shown in the study by Liu et al. [113]. Authors also revealed that after exposing mice to SWCNTs via intraperitoneal injection, the pyramidal neurons of the CA1 region were damaged and that Nissl substance loss occurred in pyramidal cells; in addition, it was found that there is an increase in the level of oxidative stress and inflammation in the brain.

Carbon-based nanomaterials are biodegradable and as for single-walled CNTs their biodegradation occurs via natural enzymatic catalysis [114]. MWCNTs undergo layer-by-layer degradation [115]. However, the safety of their use as drug delivery systems has not been proven; moreover, toxicity of carbon nanotubes was shown in several model test systems [116,117]. Murray et al. demonstrated that single-walled CNTs (SWCNTs) could induce dermal toxicity via oxidative stress and inflammation [118].

CNTs are classified as High Aspect Ratio Nanomaterials (HARN) due to at least one dimension of them less than 100 nm [119]. So CNTs can be considered as fibers, their potential adverse effect can be related to the capacity of these materials to induce oxidative stress in cells [120]. The presence of CNTs, near or inside cells, may lead to ROS production and to an overloading of the cell antioxidant defense system. This phenomenon can induce toxic effects such as alteration of DNA (genotoxic effects) which can ultimately lead to tumor development.

The study of Urankar et al. demonstrated the indirect development of toxicity and the presence of complex mechanisms of their manifestation. Thus, oropharyngeal aspiration of Multi-walled CNTs (MWCNTs) increased the susceptibility of cardiac tissue to ischemia/reperfusion injury without significant inflammation in the airways [121].

In general, the toxicity of carbon nanomaterials upon penetration into cells depends on their physical properties, with length playing a particularly important role. Thus, with prolonged exposure to lenghty carbon nanotubes, there is a tendency to develop inflammation and fibrosis [122] Moreover such factors as concentration/dose of CNTs, number of layers in nanotubes, catalyst residues left over during synthesis or functionalization, degree of aggregation, oxidisation and functionalisation have been found to have an influence on the degree of toxicity of CNTs [123]. However, the influence of metal ion residues during the industrial formation of carbon nanotubes is debatable. Thurnherr et al. [123] and Aldieri et al. [124], for example, noted that impurities affect the level of toxicity, while there is an opposite opinion from other researchers [125–127] Therefore, more detailed research and development of functional coatings and modifications are still needed that will reduce the negative effect.

### 3.2. Clay Nanotubes-Based Drug Delivery Systems

In recent years, the interest of a number of researchers in the field of the formation of drug delivery systems has been directed to the potential of using the natural halloysite aluminosilicate as nanocontainers for loading various drugs. Loading of different compounds

of various nature into the lumens of halloysite nanotubes (HNTs) can be easily achieved via vacuum-facilitated loading (Figure 7). Halloysite nanotubes are considered advantageous over other nanotubes, e.g., carbon nanotubes or inorganic nanotubes made of tungsten or titanium, etc. Halloysite is a hollow tubular structure, up to 1 μm long, with an outer diameter of up to 70 nm and an inner lumen of up to 15 nm [128,129]. The outer surface of HNTs is negatively charged, while the inner cavity carries a predominantly positive charge. Due to the presence of oppositely charged inner cavity and outer surface, selective modification of the inner and outer surfaces of halloysite by different molecules can be achieved [130].

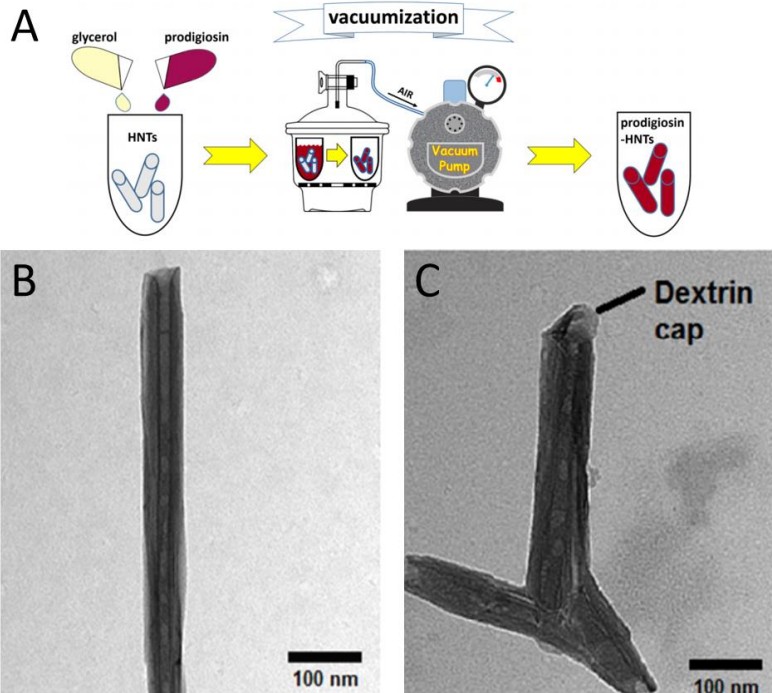

**Figure 7.** Loading of drugs into HNTs lumen. (**A**)—Scheme of vacuum-facilitated loading of prodigiosin (Adapted from [131]), (**B**,**C**)—TEM images of HNTs loaded with paclitaxel without dextrin caps (**B**) and with dextrin caps (**C**). Adapted from [132] with permissions from Elsevier Inc.©.

The main advantage of HNTs is low toxicity shown in various test systems [133–139], the possibility of modifying the internal cavity to increase loading efficiency by etching the inner cavity of the nanotubes by treating HNTs with strong acids to chemically attack the alumina sheets on the inner surface [140–142]. HNTs were successfully used for the hair coating on the base of self-assembly mechanism [143]. HNTs-based fluorescent composites have been studied as stable and long-lasting biological markers, enabling halloysite visualization in biological objects and new target-delivery materials [144]. HNTs modified with biocompatible fluorescent tags may be suitable for better understanding the drug transport to target tissues [145].

Thus, the absence of toxicity, physicochemical properties, and the possibility of modification make halloysite nanotubes a promising candidate for the formation of nanocontainers used for targeted drug delivery of anticancer natural medication, like curcumin and prodigiosin [131,146,147] HNTs were also proven to be a unique and promising vehicle for the fabrication of DNA and RNA delivery systems. For example in work of Shi et al. HNTs covered with 3-amino-propyl-triethoxysilane (APTES) were given as an antisense oligonucleotide to the HeLa cells [148]. Some researches describe the systematic investigations of HNTs to fabricate the oral drug delivery platform on the base of commercially available drugs including resveratrol, paclitaxel, verapamil [132,149–156]. Recently, Lazzara et al. conducted an extensive in vitro toxicity and genotoxicity analysis of halloysite nanotubes from various sources in HeLa, CHO and HepG2 cell lines and concluded that halloysite has good biocompatibility at low doses and short exposure times [157]. Oppositely work of Toledano-Magaña showed that HNTs and HNTs functionalized with polystyrene did

not substantially affect the viability of human and mice macrophages at concentrations as high as 100 μg mL$^{-1}$ which concidered as relatively high. At this concentration, FHNT induces the release of some proinflammatory cytokines in these cells, followed by the consequent production of anti-inflammatory cytokines, probably as physiological feedback for controlling inflammation. The cytokines levels induced by FHNTs were consistently lower than that of HNTs and, in turn, much quieter than those caused by talc, considered as inert material [158].

The main problem of targeted drug delivery is the controlled release of the loaded drug at a targeted locus. The proposed solutions to this problem are associated with the formation of additional functional coatings that respond to external and internal stimuli, such as pH, temperature changes [159–161]. Another approach to control the release of loaded compounds is the formation of functional plugs at the ends of tubular nanocontainers [162]. The disadvantage of this approach is the use of aggressive conditions to activate the stoppers. In this regard, an approach was proposed that involves the formation of coatings and end stoppers activated upon exposure to certain intracellular enzymes [21,23]. In such systems, the release of the loaded drug occurs only when a certain intracellular enzyme is activated. In particular, systems based on HNTs and dextrin loaded with a drug were obtained [163]; the release of drug occurred in the presence of an intracellular enzyme that decomposes polysaccharides (Figure 8).

However, despite active studies of halloysite nanotubes as drug delivery systems, the question of the possibility of HNTs using for intravenous drug delivery remains open. The most possible application of halloysite nanotubes seems to be their use as a topical agent for the treatment of skin diseases, as well as in the intestinal lumen as a drug carrier in the form of an orally administered drug.

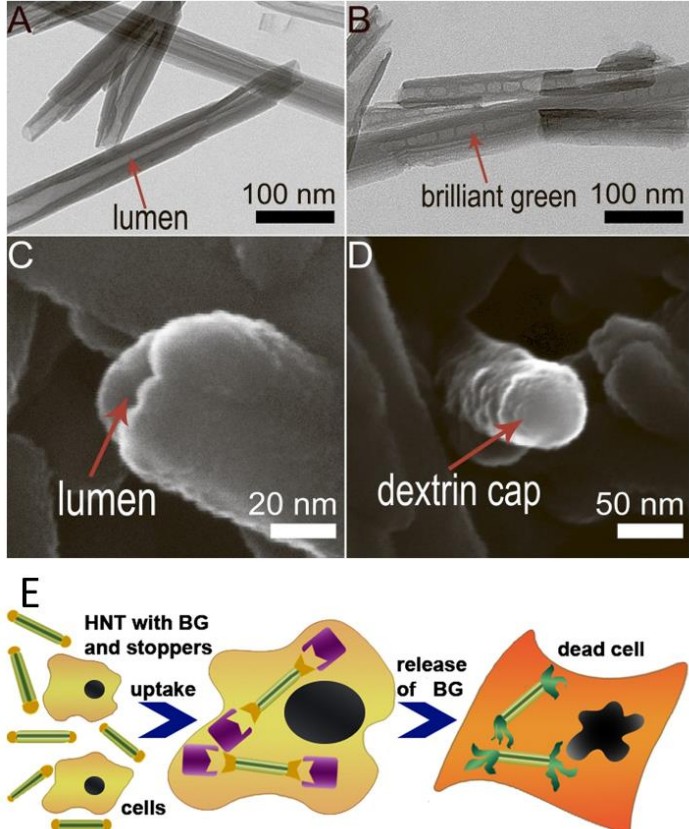

**Figure 8.** (**A**)—TEM image of pristine HNTs; (**B**)—TEM image of brilliant green loaded halloysite; (**C**)—SEM image of an end of pristine nanotube with open lumen; (**D**)—SEM image of a dextrin cap on the end of the nanotube; (**E**)—a scheme of the enzyme-activated anticancer drug delivery system based on drug-loaded HNTs with dextrin stoppers. Reproduced with permission from [163].

## 4. Dendrimers as Anti-Cancer Drug Delivery System

Dendrimers are highly branched, three-dimensional macromolecules with well-defined structures (Figure 9). Their unique architecture allows for precise control over size, shape, and surface functionality, making them incredibly versatile for various applications. In drug delivery, dendrimers can be used to encapsulate drugs within their structure, protect them from degradation, and deliver them to specific targets in the body [164].

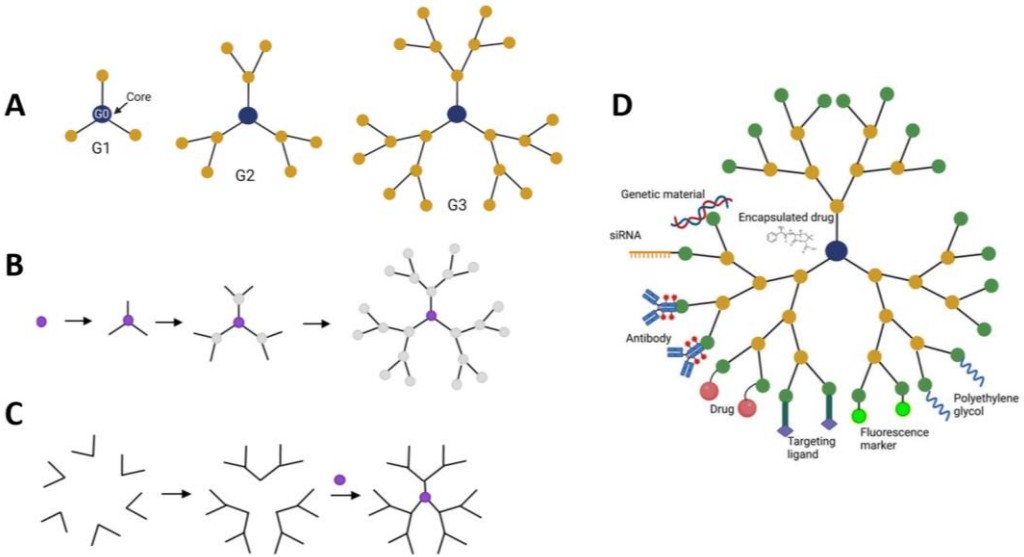

**Figure 9.** Schematics representation of dendrimers, methods of fabrication and modifications. (**A**)—dendrimers with one, two, and three generations (G1, G2 and G3). Each generation creates new moieties for attachment of functional ligands and therapeutics; (**B**)—the divergent method of dendrimer synthesis; (**C**)—dendrimer modifications and therapeutic attachments. (**A,D**) adapted from [165] under an open access Creative Commons CC BY 4.0 license; (**B,C**) adapted from [164].

When used as drug delivery vehicles, dendrimers can improve the solubility, stability, and bioavailability of drugs. Their surface functional groups can be modified to enhance targeting capabilities, allowing for site-specific delivery of drugs to tissues or cells of interest. Additionally, dendrimers can be engineered to release drugs in a controlled manner, prolonging their therapeutic effects and reducing side effects [166]. Dendrimers can improve the solubility of poorly soluble drugs, increasing their bioavailability and therapeutic efficacy and can release drugs in a controlled manner, ensuring sustained drug concentrations at the target site.

Due to the ability to regulate the size of molecules, PAMAM dendrimers used as drug carriers can be used in anticancer therapy. In particular, Yao and Ma synthesized a complex of biotinylated PAMAM dendrimer with the anticancer drug paclitaxel and demonstrated an increase in its cellular uptake by cancer cells followed by a decrease in its cytotoxicity effects [167].

Lysine-based dendrimers such as PLL have also found use in biomedical research. They have a high level of biocompatibility as well as good biodegradability, and they also contain many amino functional groups that allow drug conjugation [168].

They also have antiangiogenic activity, shown in a number of studies, which promotes the death of tumor cells, delays tumor growth and reduces toxicity to healthy tissues [169,170].

Polypropylene imine (PPI) dendrimers are a third type of dendrimers commonly investigated for biomedical purposes [171]. Propylene imine monomers form the branching units of each generation. They have potential in fabrication of anti-cancer drug delivery systems due to theit cationic charges can destabilize negatively charged tumor cells membranes and promote drug penetration [169].

Drugs can be encapsulated within the inner core or attached to surface functional groups. Dendrimers can carry different types of cargo, including DNA, siRNA, antibodies,

and drugs. Modification with polyethlene glycol can improve dendrimer "stealth" and bioavailability. Moreover, fluorescence markers can be attached for drug tracking. To improve targeting and minimizing side-effects, specific ligands can be attached to the dendrimer surface to be recognized by receptors overexpressed by cancer cells [167].

As researchers continue to explore the potential of dendrimers in drug delivery, several challenges remain to be addressed. These include optimizing dendrimer synthesis methods, improving biocompatibility, and ensuring regulatory approval for clinical use. Despite these challenges, the unique properties of dendrimers hold great promise for revolutionizing drug delivery and personalized medicine.

## 5. Micelles and Liposomes as Anti-Cancer Drug Delivery Vehicles

Liposomes are the earliest nanostructured carriers that are widely used in cosmetology and biomedicine. The development of this line of research began since the end of 1960s of the XX century from the works of Weissman devoted to the immobilization of enzymes [172]. The global market of liposomes is constantly growing. And according to market research companies, the liposome drug delivery market was worth $4.7 billion in 2022, with a projected volume of $10 billion by 2031 [173].

Liposomes, microscopic hollow spheres ranging in size from 50 to 1000 nm, composed of lipids, mimic the structure of cell membranes made by the self-assembly of diacyl-chain phospholipids (lipid bilayer) in aqueous solutions while the polar "heads" of the phospholipid are facing the solvent, and the tail parts are facing to each other, thus forming an internal water space or core [174]. This unique characteristic enables liposomes to encapsulate drugs and deliver them directly to cancer cells. By leveraging the body's own biological processes, liposomes enhance drug efficacy and reduce toxicity of anti-tumor drugs, opening a new frontier in cancer therapy [175,176].

From the traditional thin film hydration method to the advanced microfluidic method, researchers have a variety of techniques at their disposal for preparing liposomes with specific properties. The fabrication of liposomes involves several methods, each with its own advantages and disadvantages which were described in details in reviews [177,178].

The thin film hydration method (Figure 10) which also called Bangham method, is one of the earliest and most widely used methods for liposome fabrication. In this method, lipid components are dissolved in an organic solvent to form a thin lipid film on the walls of a round-bottom flask. The film is then hydrated with an aqueous solution, leading to the formation of multilamellar vesicles (MLVs). These MLVs can be further processed into small unilamellar vesicles (SUVs) or large unilamellar vesicles (LUVs) through extrusion or sonication [179].

In this method lipophilic drugs can be dissolved with the phospholipids mixture prior to the thin film formation; hydrophilic cargoes can be inserted within the hydration mediums and then passively incorporated into the liposome during the hydration process [177].

The reverse phase evaporation method is another common technique for liposome fabrication, particularly for the encapsulation of hydrophobic drugs. In this method, lipid components and the drug of interest are dissolved in an organic solvent to form a water-in-oil emulsion (Figure 11). The organic solvent is then evaporated under reduced pressure, leading to the formation of multivesicular liposomes (MVLs). These MVLs can be further processed into smaller vesicles through extrusion or sonication.

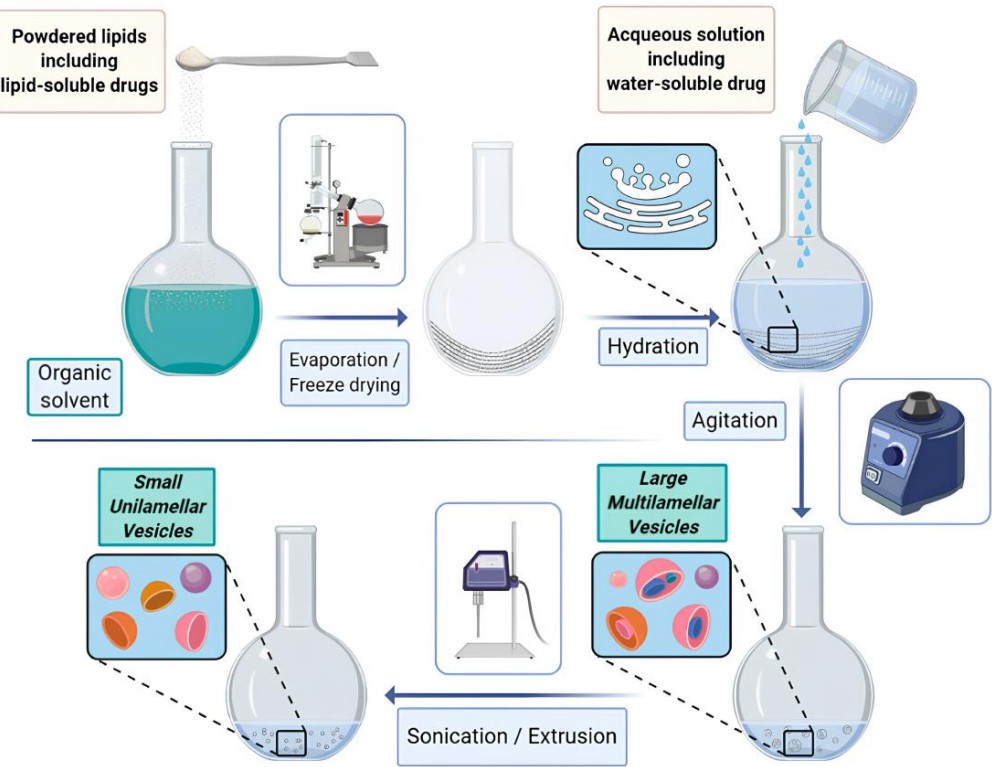

**Figure 10.** Scheme of the thin-film hydration method of liposome preparation. From [179] under an open access Creative Commons CC BY 4.0 license.

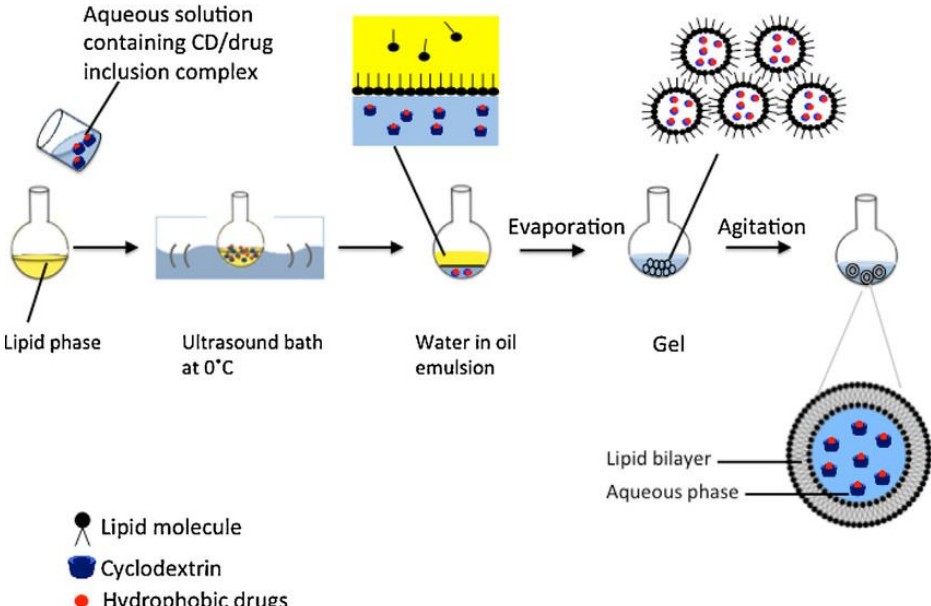

**Figure 11.** Scheme of the reverse phase evaporation method of liposome preparation. Reproduced with permission from [180].

Traditional methods of liposome fabrication often result in polydispersed liposome populations with varying sizes, leading to inconsistencies in drug delivery efficacy. In general, drug delivery nanocarriers with a consistent and narrow size distribution are essential to achieve optimal clinical results [181]. Microfluidic methods involve the controlled manipulation of fluids in microscale channels, allowing for the production of monodisperse liposomes with uniform size and shape (Figure 12).

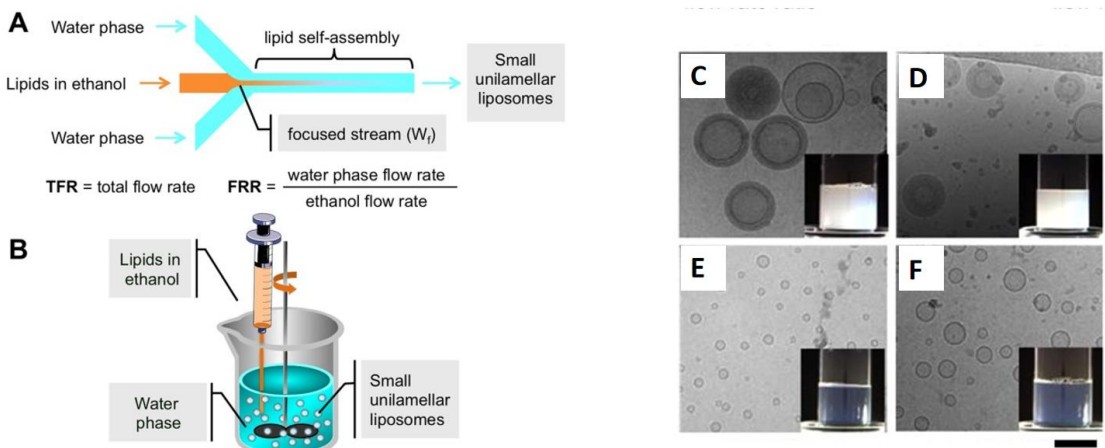

**Figure 12.** (**A**)Schematic representation of a microfluidic device, namely a MHF microchip (#chip1-MHF), and the process of liposome (SUV) self-assembly; (**B**)—schematic representation of the ethanol injection procedure; Cryo-TEM images and macroscopic aspect (insets) of empty PC/cholesterol (**C**) and PC/DDAB (**E**) are reported. For comparison, images of the corresponding ivermectin loaded liposomes are also reported (**D**,**F**). Bar corresponds to 100 nm. Adapted from [182] under an open access Creative Commons CC BY 4.0 license.

Hydrophilic core of liposomes is most suitable for encapsulating water-soluble hydrophilic agents, while hydrophobic, lipid-soluble (hydrophobic) and amphiphilic compounds can be included into lipid bilayer (Figure 13) [183].

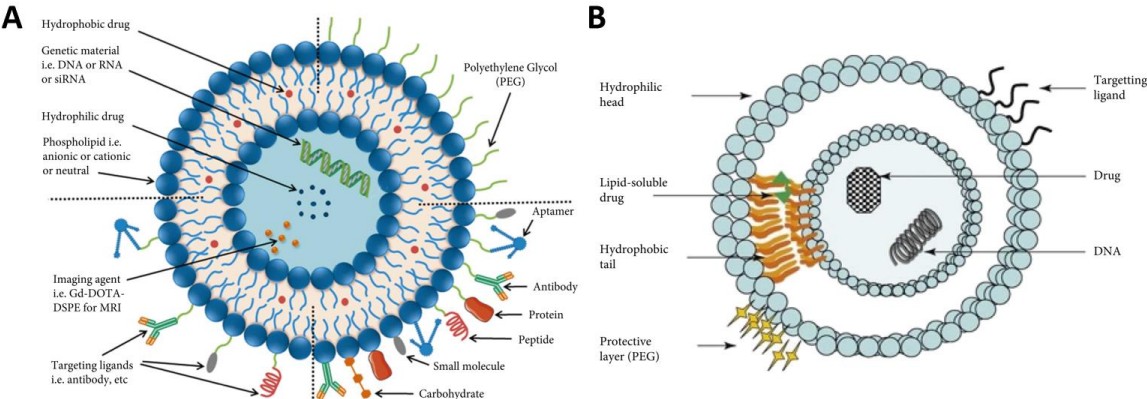

**Figure 13.** Surface functionalized liposomes as drug-delivery system (**A**): Scheme of Liposome various targeting ligands for enhanced delivery of payload at tumor site. Reproduced from [184] under an open access Creative Commons CC BY 4.0 license. (**B**): Liposome entrapping both hydrophilic and hydrophobic drugs in the aqueous core and lipid bilayer, respectively. +e surface of the liposome allows for the addition of targeting ligands and a polyethylene glycol coating for active and passive targeting, respectively. Reproduced from [185] with permissions from Elsevier Inc.©.

Liposomes allow for combination therapy, where multiple drugs can be encapsulated within a single vesicle, synergistically targeting different aspects of tumor growth. Liposomes act as carriers, ensuring the safe delivery of anticancer drugs to tumor sites while minimizing damage to healthy tissue, moreover they are biodegradable and do not cause an immune response. When liposomes encounter cancer cells, they exploit the phenomenon of enhanced permeability and retention (EPR) [186], allowing them to selectively accumulate in the tumor tissue. Once inside the tumor, liposomes release the encapsulated drugs, targeting and killing the malignant cells. In addition to the main delivery systems already in clinical use, such as liposome-encapsulated doxorubicin and paclitaxel, the development of new drugs continues to address emerging medical challenges. Thus, in 2021, lyophilized

remdesivir liposomes were developed with the possibility to reconstitute into liposomal aerosol for the treatment of infection caused by COVID-19. The developed system made it possible to improve the in vivo behavior of existing remdesivir cyclodextrin conclusion compound injections. In addition, liposome encapsulation endowed remdesivir with much higher solubility and better biocompatibility [187].

Liposomes have shown tremendous potential in revolutionizing cancer treatment and offer several advantages over traditional cancer therapies, making them an attractive option for patients and healthcare providers alike. Here are some key benefits of utilizing liposomes in cancer treatment:

- Enhanced drug delivery: liposomes improve drug solubility and stability, enabling efficient delivery of therapeutic agents.
- Targeted therapy: by selectively delivering drugs to tumor sites, liposomes minimize damage to healthy cells and reduce side effects.
- Controlled drug release: liposomes can be engineered to release drugs in a controlled manner, ensuring sustained therapeutic levels within the tumor.
- Overcoming drug resistance: liposomal formulations can overcome multidrug resistance mechanisms, enhancing the effectiveness of chemotherapy.
- Combination therapy: liposomes allow for combination therapy, where multiple drugs can be encapsulated within a single vesicle, synergistically targeting different aspects of tumor growth.

Numerous clinical trials are underway worldwide to evaluate the safety and efficacy of liposomal therapies for various types of cancer (Figure 14). Overall, more than 1700 clinical studies of liposomes as anticancer drug delivery systems have been reported, but 470 of them are currently active [188]. These trials aim to validate the potential of liposomes and pave the way for their wider adoption.

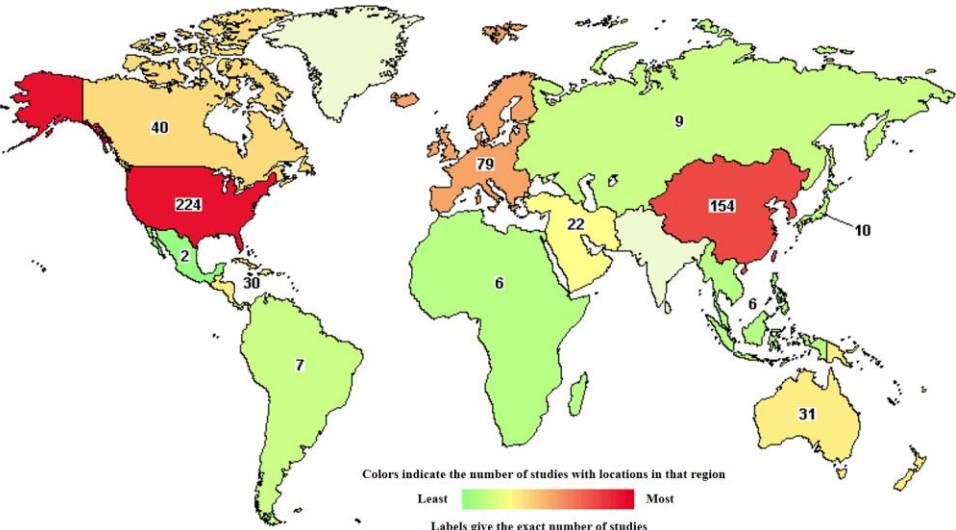

**Figure 14.** Worldwide distribution of active clinical trials of liposomal based drug-delivery systems for cancer therapy (from https://classic.clinicaltrials.gov/ct2/results/map/click?map.x=339&map.y=372&term=liposomes&recrs=abdf&cond=cancer&mapw=714) (accessed on 20 April 2024).

Clinical trials investigating liposomal doxorubicin, a widely used chemotherapy drug, have shown promising results. First registered drug Doxil® or Caelyx® was presented in 1995 by Sequus Pharmaceuticals. Doxil was designed as a polyethylene glycol coated doxorubicin (DOX) liposome intended for the treatment of Kaposi's sarcoma [189]. Notably, liposomal formulations have demonstrated enhanced antitumor activity and reduced cardiac toxicity compared to conventional doxorubicin [190]. Another commercial formulation Myocet® is a non-PEGylated liposomes encapsulating doxorubicin also showed less cardiac side effects and a shorter circulation half-life [191,192].

Paclitaxel is a naturally derived substance that was originally extracted from the pacific yew tree Taxus brevifolia. Paclitaxel is a relatively new anti-microtubule drug that gained popularity due to its anticancer activity and commonly used for different types of cancers treatment, including breast cancer, non-small cell lung cancer, gastric cancer, head and neck cancer and other [193]. Liposomal encapsulated paclitaxel still not approved by FDA, but some studies and clinical trials have revealed improved response rates and prolonged survival in cancer patients treated with liposomal paclitaxel. In study of Wu with co-workers the improved efficacy and less side-effects as compared to common paclitaxel have been demonstrated [194]. Paclitaxel was also incorporated into Lipusu® liposomes to treat gastric carcinoma efficiently with much less adverse effects [195]. The liposomal formulation also reduces the risk of severe hypersensitivity reactions associated with the conventional formulation.

Pancreatic cancer is known for its aggressive nature and limited treatment options. Liposomal irinotecan, a prodrug formulation, has shown promise in clinical trials for pancreatic cancer. The liposomal delivery system improves drug stability and bioavailability, leading to improved patient outcomes. FDA approved formulation Onivyde® represent the PEGylated liposome carrying irinotecan and exhibits a long-acting, antitumor effect [196].

In addition to above mentioned drugs, there are reports on the development and analysis of liposomal delivery systems for different anti-cancer drugs such as daunorubicin (Daunoxome) and annamycin (Annamycin); platinum compounds—cisplatin (Lipoplatin, Platar); retinoic acid—tretinoin (ATRA-VI) and altragen; alkaloid vincristine [197–200]. A new liposomes formulation called Mepact® based on Muramyl Tripeptide-Phosphatidyl Ethanolamine was globally approved for the treatment of osteosarcoma [201]. Moreover aptamer-functionalized liposomal Fluorouracil 5-FU (AFL5-FU) coated by calcium alginate/chitosan/PEC reported to provide an effective oral formulation for colon-cancer therapy. In the study of Khodarahmi et al. was demonstrated that nanoencapsulation of the synthesized liposomes with calcium alginate/chitosan complex promote the targeted and selective drug delivery to colon cancer cells in vitro [202]. Cadinoiu et al. demonstrated that crosslinked gel based on sodium alginate and hyaluronic acid containing AS1411-aptamer conjugated liposomes loaded with 5-fluorouracil can be used as a new therapeutic approach for the topical treatment of basal cell carcinoma [203].

Micelles are constructed from amphiphilic block copolymers, which spontaneously form core-shell aggregates when a critical micelle concentration is reached (Figure 15) [204]. Unlike liposomes, micelles can load hydrophobic drugs into the core and stabilize in an aqueous environment with good stability. Drugs are absorbed by them in three different ways: physical capture, chemical conjugation and electrostatic effect. For targeted drug delivery, the surface of micelles can also be decorated with various ligands.

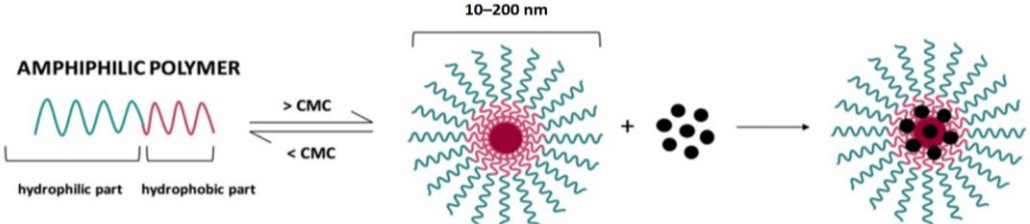

**Figure 15.** Schematic representation of micelles. Reproduced from [204] under an open access Creative Commons CC BY 4.0 license.

However, the low stability of micelles in vivo limits their further development as drug carriers. Studies have shown that drug-loaded micelles degrade into free surfactants and drugs with a significant reduction in bioavailability and therapeutic properties. The stability of micelles can be improved using physicochemical strategies, but these methods can complicate the large-scale production of micelles. Two approaches are under investigation currently to combine micelles and liposomes: one is to attach charged micelles to charged liposomes by electrostatic effect, and the second is to encapsulate micelles with liposomes

in the inner aqueous phase (Figure 16). These methods are thoroughly described in the review of Qian et al. [205].

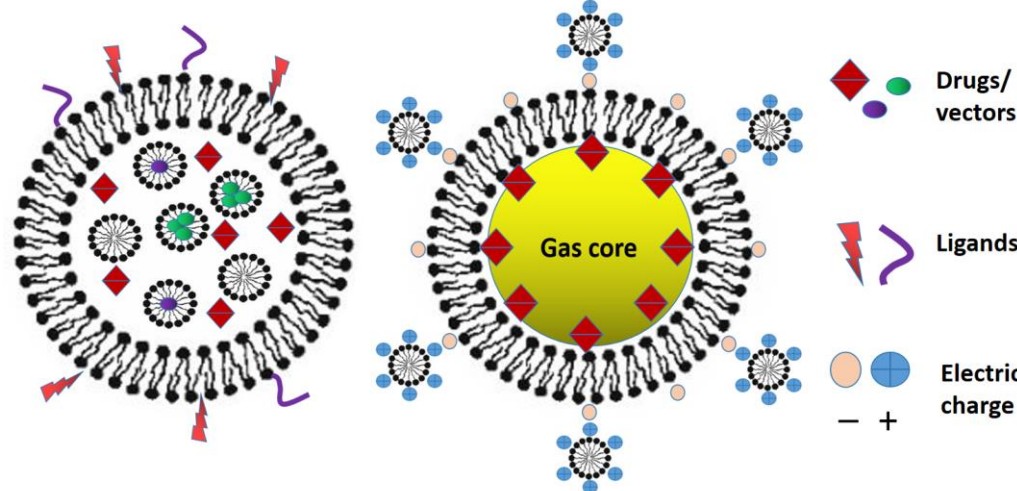

**Figure 16.** Two approaches of liposomes and micelles combination.

Therefore, liposomes and micelles offer a unique and promising approach to cancer treatment. Their ability to deliver drugs directly to tumor sites, reduce systemic toxicity, and offer sustained release has the potential to revolutionize the field of oncology. Ongoing research and clinical trials of various liposome-based delivery system options may ultimately lead to the development of unique complex formulations for effective cancer treatment. Moreover, combination of liposomes and micelles can lead to the creation of unique drug delivery systems due to the combination of their properties.

## 6. Comparative Analysis of Different Drug Delivery Systems, Their Advantages and Disadvantages

The main requirement for therapeutic nanoparticles and nanomaterials is their biocompatibility and safety. It is obvious that research of long-term safety/toxicity of the nanoparticles, polymer nanocomposites and also products of their modification in the organisms and components after destruction in case of their bioassimilation are more informative [206].

When testing the biosafety of nanomaterials (NMs), they are often limited to test objects of low levels of biological organization, such as animal cells, protists, nematodes. However, some risks of using NMs can be elucidated only in systems such as organs or whole organisms. Even detailed clinical studies do not exclude individual negative reactions, such as contact sensitivity, allergies, as well as delayed consequences after prolonged nanotherapy.

Gold nanoparticles (AuNP) can be synthesized using different approaches into a variety of shapes (spheres, rods, shells, cubic nanocages and clusters), sizes (typically range from 1 to 100 nm) and respectively physical, chemical and biological properties [207]. The listed parameters had much influence on biodistribution, while AuNP surface coating plays an important role in toxicity [208]. AuNPs are a promising tool for transdermal drug delivery, due to their unique properties and surface adjustability.

AuNPs are highly biocompatible and safety materials that has also been proven by clinical trials [209] however, the number of this clinical studies and total patient number remains limited. Indirect toxic effects of AuNP expressed through oxidative damage to cell lines in vitro and in liver, spleen and kidney in vivo [210]. In addition, many studies have demonstrated their potential to modulate various immune cell activity, so allergenicity of AuNP remains unclear [211]. In Local Lymph Node Assay (LLNA) in mice, soluble gold (III) chloride (AuCl3) caused lymph node expansion (SI 10.9), whereas bulk particles (Au,

942 nm) and AuNP did not [211]. Apparently, the immunogenicity of gold depends on solubility and is not detected with metallic and insoluble forms as AuNPs [212].

Like gold nanoparticles, iron oxide MNPs (IONPs) have been widely used in Biomedicine, including drug delivery, magnetic resonance imaging, anticancer hyperthermia therapy (anticancer treatment/hyperthermia treatment for cancer), but the key difference in the possibility of magnetically controlled guidance and accumulation of MNPs in specific site. IONPs readily pass the blood brain barrier (BBB) and convenient for the measurement of the BBB leakiness and magnetic drug targeting to brain tumors [213]. The penetration of NPs through tissue barriers depends primarily on their size, that typically for iron oxide MNPs (IONPs) in range from 5 to 20 nm. Particles below 15 nm have the short blood circulation time and are rapidly removed through eructation and renal clearance [214] have limited application when prolonged action is required, but quite promising in cosmetology for transdermal drug delivery [215].

Although there is evidence that Iron oxide MNPs (IONPs) participate in the production of oxidative stress that leads to cell damage [216], IONPs themselves are non-toxic and easily removed from cells and tissues but lose their superparamagnetic properties over time or suffer chemical degradation in bioenvironment. Iron oxide MNPs (IONPs) are by far the most studied MNPs whose safety has been proven in many studies. Increasing the stability of IONPs may be harmful for biomedical use since when a significant amount of IONPs (for rats treated with peritoneal injections concentrations more than 1.7 mL/kg) is introduced into the body, an accumulation effect occurs with negative consequences [217].

One of strategy for changes of the magnetic properties and bioactivity includes doping IONPs with other metals, such as cobalt (Co), zinc (Zn), manganese (Mn), and gadolinium (Gd) [218] resulting in the fabrication of unique multimodal nanoparticles [219]. Several metal-doped IONPs exhibit antimicrobial activity through generation of ROS [220] and, thus, potential toxic.

$CoFe_2O_4$ nanoparticles have been shown to increase the production of ROS in many in vitro and in vivo toxicological studies [221]. Obviously the inclusion of a second metal must not induce additional toxicity, as it will not have any application in pharmacology. The use of coatings that prevent metal ion leakage is a common effective approach for biocompatibility and limiting toxicity [222].

Polymeric coatings such as alginate, chitosan, dextran, polyethyleneimine (PEI), poly(vinyl alcohol) (PVA), poly(ethylene glycol) (PEG), poly(lactic-co-glycolic acid) (PLGA), and its copolymers can facilitate maintain colloidal stability of NPs and dispersibility, which is important for biomedical researches. PEGylated and dextran coated nanoparticles are approved as either new drug applications and clinical use [223,224].

The multifunctional nanocomplexes with PEI enhanced transfection while decreased PEI toxicity and escape lysosomal degradation following internalization [225].

The use of various copolymers, antibodies, targeting ligands, and inorganic compounds (metals, graphene, silica) creates a wide variety of NP properties. IONPs with radioisotopes onto the surface is emerging as a novel tool for molecular imaging. Functionalization with coatings changes the biodistribution of NPs, firstly, due to the different affinities of the coatings (PEG, PLGA), target specificity (antibodies, ligands) and also as a result of the hydrodynamic diameter of the NPs increase [226].

In some cases, coatings produce mutually exclusive effects, for example, MNPs nanocomposites with silica can specifically bind to nucleic acids; with gold or silver allowing facilitate organic conjugation, biocompatibility and protecting MNPs from oxidation, but weaken the magnetic properties [227].

Carbon nanotubes (CNTs) can be advantageous for biological imaging applications as fluorescent probes with superior resolution [228]. Moreover, compared to fluorescent nanomaterials synthesized from heavy metals, CNMs are biocompatible, and their surface and ends can be functionalized to fine-tune their biointerfacial properties.

CNTs have been used to contrast images of intact tissues at lower excitation powers and map the tumor region, which can be coupled with the delivery of antitumor therapeutic

agents. The stable photoluminescence of CNTs allows long-term tracking (up to hours) the motion of kinesins (intracellular motor proteins), organelles including chloroplast and mitochondria in live cells [228].

The diameter of CNTs is typically single-nanometer-order-size; their length could extend for up to ~1 μm and their loading capacity is not as high compared to halloysite nanotubes. Does this mean that to achieve a therapeutic concentration of a drug when using NT as a carrier, more CNTs is required than HNPs still unclear.

The needle-like shape of the CNTs allows them to perforate cellular membranes and deliver the carried therapeutic molecules to the cellular compartments [229]. CNTs have shown benefits as a tool for small molecule drug delivery, nucleic acids and therapeutic proteins for regenerative medicine [230]. CNTs can be covalently modified with a variety drugs or adsorb aromatic drugs via hydrophobic interactions with aromatic surface of the CNTs [231]. However, the inherently hydrophobic nature of CNTs can limit loading efficiency of water-soluble drugs and dispersibility CNTs in aqueous environments. The potential of CNTs for drug loading can be increased with different polymers, for example, poly-ε-caprolactone (PCL) and other, grafted to the surfaces of CNTs [231]. The surface functionalisation and coating is often utilised to improve the colloidal stability and bio-compatibility, reduction cytotoxicity of CNTs. At the same time, it is necessary to take into account that the biological response to nanocomposites depends on a number of factors, including the amount of nanomaterials, the concentration of substances, and the complexity of the interacted biosystem.

HNTs are available at the scale of thousands of tons (abundant natural reserves) and accordingly, much cheaper compared for CNTs [232]. Native HNTs with the negative Si–O–Si groups on the outer surface and the positive aluminol (Al-OH) groups on the inner surface can adsorb both cationic and anionic molecules.

The surface of HNTs is suitable for the immobilization of metal nanoparticles, including MNPs, which allows the targeting of the HNTs-MNPs nanocomplex using a magnetic field. However, to immobilize some metals, additional functionalization of the HNTs surface is required [233]. The positively charged environment inside the lumen of halloysite nanotubes (HNTs) were utilized to synthesize nanoparticles in situ within the HNTs [234] in order fabricating nanocomposite that selectively acts on cancer cells during chemodynamic therapy.

It had been assessed that pure halloysite is not toxic for the cells, but in mice via the oral route (from 50 to 300 mg·kg$^{-1}$ BW) HNTs' toxicity effects was observed. It could be associated with HNTs accumulation in the mouse liver after 30 days of prolonged administration [233]. Thus, when fabricated nanoformulations with halloysite, it is necessary to take into account that this bioinert and biocompatible mineral is not biodegradable.

HNTs have a certain penetrate ability to the cell membrane through both clathrin-dependent and independent endocytosis and are subsequently localized in the perinuclear region surrounding the cell nuclei [235]. The microtubules which can become targets for the action of substances loaded into HNTs, also participate in the halloysite cellular trafficking. There is no data in the literature yet on the ability of halloysite nanotubes to pass through skin barriers; probably the safest way to use nanotubes is the surface treatment of skin or its derivatives—hair for some medicinal and cosmetic purposes [215].

HNTs are a suitable delivery vehicle for water-insoluble natural anticancer drugs such as resveratrol [155] and prodigiosin. Prodigiosin, a poor water soluble drug, was loaded onto HNTs from glycerin based solvent containing ethanol, which provided prodigiosin release from nanotubes into cells cytoplasm and the absence of extracellular leakage of prodigiosin [131].

Nanotubes are thermostable containers that prevent thermal destruction of loaded thermolabile proteins. One promising area is the immobilization of enzymes on the surface and in the lumen of HNTs. Depending on the enzymes charge, different interaction sites on HNTs and subsequently their release from the halloysite lumen over time are observed. As an example, glucose oxidase entrapped into HNTs showed an improved thermal stability as well as an activity of storage time. Different adsorbed enzymes (laccase, glucose oxidase,

lipase, and pepsin) exhibited improved biocatalytic abilities depending on pH conditions in tests [236].

It is difficult to obtain stable dispersions of HNTs in water and thus usability for its practical application without the use of coatings due to the size of HNTs. In composites HNTs-polymers, halloysite provide mechanical strength and rigidity, biocompatibility, swelling properties and colloidal stability. Polymer coatings was also useful for the sustained release of drugs from HNTs. The degree of dispersion of HNTs and the interfacial interactions between polymers and HNTs (electrostatic and hydrogen bonding interactions) are crucial factors affecting the performance of composites. And since halloysite is not biodegradable, bioinert, the release of substances is controlled only by solubility in water, the presence and type of stoppers and surface coatings [233].

Liposomal-based drug delivery systems can encapsulate both hydrophobic and hydrophilic drugs and provide controlled distribution and sustained release. The surface of liposomes is easily modified for targeted, prolonged and site specific therapeutic action [237]. Unlike other nanosized delivery systems, liposomal-based drug delivery systems reached advanced phases in clinical trials a significant part of them are currently clinically approved to treat several diseases, such as cancer, fungal and viral infections. In addition, cationic liposome-DNA complexes (lipoplexes) are now considered as a potentially viable alternative to viral vectors for the delivery of therapeutic genes [238].

In general, comparative analysis of various nanoscale delivery systems requires compliance with a number of conditions. In particular, parameters such as the amount of nanocarriers based on their loading capacity to achieve effective therapeutic drug concentrations, drug release conditions, and testing systems must be investigated on a single platform, keeping all experimental conditions constant.

A similar approach was used in a comparative analysis several nanocarriers (dendrimer, liposomes, carbon nanotubes, Poly(d,l-lactide-co-glycolide) (PLGA) nanoparticles) for their anticancer drug (docetaxel, DTX) delivery potential by comparing them on similar ground of parameters like optimal drug loading efficiency, drug release, hemolytic toxicity, anticancer potential, etc. [239]. From the outcomes this study it can be concluded that carbon nanotubes have the best parameters (biocompatibility, drug release profile) for DTX nanodelivery.

Every nanosystem has its own advantages and contributes in the development of an effective drug delivery methods. A clear way of comparison is to use one drug in different delivery nanosystems in one platform assessment to eliminate differences in conditions. However, due to the fact that all systems differ according to the above mentioned criteria, an assessment of the feasibility of their use in each specific case should be carried out taking into account existing experience.

## 7. Conclusions

Thus, nanostructured drug delivery systems are rapidly developing. The study of nanomedicines is now at various stages of their preclinical and clinical trials, while the search for new formulations tends to continue. This is facilitated by the nature of cancer itself, which is very complex; the molecular pathways for the development of drug resistance and the mechanisms of its spread in the body are not fully understood. The trend in the design of new nano-platforms for the treatment of cancer is seen in the creation of bioengineered targeted formulations with minimal side effects in relation to healthy tissues, as well as in the formation of personalized drugs. In general, nanostructured drug delivery systems help solve problems associated with the use of traditional chemotherapeutic agents, including nonspecific biodistribution, lack of water solubility, and low bioavailability. Each type of nanomaterial offers distinct benefits in terms of drug encapsulation, target specificity, and controlled release mechanisms.

Due to some of the unique properties of nanoparticles and complexes that have been discussed in this review, the future of cancer therapy looks promising, with the potential to save lives and alleviate the burden of this devastating disease.

**Author Contributions:** Conceptualization, E.N., I.G. and M.G.; methodology, E.N. and M.G.; writing—original draft preparation, E.N.; writing—review and editing, E.N., M.G. and I.G.; project administration, M.G. All authors have read and agreed to the published version of the manuscript.

**Funding:** The reported study was funded by RSF according to the research project No. 23-25-00334.

**Data Availability Statement:** All data generated or analyzed during this study are included in this published article.

**Acknowledgments:** This paper has been supported by the Kazan Federal University Strategic Academic Leadership Program (PRIORITY-2030).

**Conflicts of Interest:** The authors declare no conflict of interest.

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
