# Peer review of "Drug Delivery Nano-Platforms for Advanced Cancer Therapy"

_scipharm, doi:10.3390/scipharm92020028_

Round 1
Reviewer 1 Report
Comments and Suggestions for Authors
Review report on the manuscript titled “Drug delivery nano-platforms for advanced cancer therapy” The manuscript is well organized but it could be improved in terms of contents and quality of presentation. This reviewer suggests the publication after some minor revisions listed below:
1. In the Introduction, the authors should describe more in details the currently available tools to achieve targeted and controlled drug release
2. The authors should mention and briefly describe microsystems that have emerged as useful tools for electronically controlled drug delivery (https://doi.org/10.1002/adhm.202302969)
3. In the Introduction, the main advantages of nanomaterials as carriers of drugs should be highlighted compared to standard and conventional systems
4. Which other nanomaterials and nanovectors can be used for targeted drug delivey, besides nanoparticles? This should be emphasized and elaborated more in the introduction.
5. Page 10, lines 31
6: the format should be corrected. 6. The authors should describe more in details the preparation methods for the synthesis of liposomes
7. The review would benefit from more figures that collect examples from previous works
8. The conclusions should be enriched with some guidelines for the preparation of nanomaterials for targeted and controlled drug release

Author Response
Dear Reviewer,
Thank you for the time devoted to our manuscript and valuable suggestions. We have taken the comments on board to improve and clarify the manuscript. We hope that manuscript will be now suitable for publication at the SciPharm.
Please find below a detailed point-by-point response to all your comments.
On behalf of all authors,
Ekaterina Naumenko
- In the Introduction, the authors should describe more in details the currently available tools to achieve targeted and controlled drug release
Answer: We added required information
- The authors should mention and briefly describe microsystems that have emerged as useful tools for electronically controlled drug delivery (https://doi.org/10.1002/adhm.202302969)
Answer: We added suggested reference and described this systems
- In the Introduction, the main advantages of nanomaterials as carriers of drugs should be highlighted compared to standard and conventional systems
Answer: We highlighted the differences of nano-sized vehicles
- Which other nanomaterials and nanovectors can be used for targeted drug delivey, besides nanoparticles? This should be emphasized and elaborated more in the introduction.
Answer: We added some detailes about nanomaterials used for drug delivery
- Page 10, lines 316: the format should be corrected.
Answer: Corrected
- The authors should describe more in details the preparation methods for the synthesis of Liposomes
Answer: We added requested information to the Section 4. Micelles and Liposomes as anti-cancer drug delivery vehicles
- The review would benefit from more figures that collect examples from previous works
Answer: We include some additional Figures to the manuscript
- The conclusions should be enriched with some guidelines for the preparation of nanomaterials for targeted and controlled drug release
Answer: We added additional section with comparative analysis of nano-sized drug delivery systems and how they can be modified for targeted and control release - 5. Comparative analysis of different drug delivery systems, their advantages and disadvantages
Reviewer 2 Report
Comments and Suggestions for Authors
The review submitted by Naumenko et al. try to present the recent advances in drug delivery for cancer therapy. The aim of this review is very ambitious and large. Unfortunately, the authors have not provide a real view of all the drug delivery systems used for cancer therapy. In order to be accepted, this manuscript must submit major improvements. Comments:
1. line 89: aptamers are also very important specific ligands which are successfully used for cancer therapy.
2. section 1.1: the authors must cite studies concerning the loading of iron NPs in polymer particles which are further used for targeted therapy.
3. line 315: revise
4. section 3: the authors must cite recent articles concerning the "aptamer-functionalized liposomes and loaded with 5-FU".
5. the manuscript must be completed with references concerning polymer nanoparticles functionalized with aptamers or other ligands. This is the major field of drug delivery systems used for cancer therapy and I think that a table summarizing the papers from the last 5 years can be useful.
6. micellar systems loaded with anti-tumoral drugs are another important area of research in this field.
7. finally, the authors must indicate the advantages and disadvantages of each type of drug delivery system and also, as perspectives, how they can be improved.
Author Response
Dear Reviewer,
We thank you for the time devoted to our manuscript and for the valuable notes and suggestions which help us to improve the manuscript quality and enhance its readability by introducing useful information regarding nanoscale drug delivery systems and we hope that manuscript will be now suitable for publication at the SciPharm.
Please find below a point by point reply to the comments.
On behalf of all authors,
Ekaterina Naumenko
The review submitted by Naumenko et al. try to present the recent advances in drug delivery for cancer therapy. The aim of this review is very ambitious and large. Unfortunately, the authors have not provide a real view of all the drug delivery systems used for cancer therapy. In order to be accepted, this manuscript must submit major improvements. Comments:
- line 89: aptamers are also very important specific ligands which are successfully used for cancer therapy.
We added this note in Introduction section
- section 1.1: the authors must cite studies concerning the loading of iron NPs in polymer particles which are further used for targeted therapy.
We added suggested studies in the section MNPs
- line 315: revise
Revised
- section 3: the authors must cite recent articles concerning the "aptamer-functionalized liposomes and loaded with 5-FU".
We added some references to the section 4. Micelles and Liposomes as anti-cancer drug delivery vehicles
- the manuscript must be completed with references concerning polymer nanoparticles functionalized with aptamers or other ligands. This is the major field of drug delivery systems used for cancer therapy and I think that a table summarizing the papers from the last 5 years can be useful.
We thank Reviewer for constructive suggestions . We added some references to the Introduction, but as for additional section and deep study of research works devoted to this topic we decided to take it into account in our next publication.
- micellar systems loaded with anti-tumoral drugs are another important area of research in this field.
We added brief description of main features of micellar drug delivery systems in the section 4. Micelles and Liposomes as anti-cancer drug delivery vehicles
- finally, the authors must indicate the advantages and disadvantages of each type of drug delivery system and also, as perspectives, how they can be improved.
We added additional section with comparative analysis of nano-sized drug delivery systems - 5. Comparative analysis of different drug delivery systems, their advantages and disadvantages

Reviewer 3 Report
Comments and Suggestions for Authors
Naumenko, Guryanov and Gomzikova present a comprehensive review of drug delivery systems. This subject is of high interest and this contribution may serve as a reference text. With that said I have some comments for the authors' consideration:
Minor
Just below Table 1, there is a repeated sentence.
Overall, the figures are appropriate but some of them have low resolution or illegible text. Plus, Figure 1 is cluttered with text; perhaps a redesign is due.
Major
While the presentation is very good and straightforward, I think there are some missing details that could improve the review. Such as further description of the procedures to generate the discussed nano systems.
Also, a missing aspect is a discussion or comparison between the presented methods. This would provide a wider context , while also providing perspectives on the directions in the field. On a similar note, there is little comment on the challenges or overarching difficulties associated with these platforms.
Finally, I noticed that the authors do not develop on the subject of dendrimers/polyplexes. Along with the mentioned platforms these also hold promise on the development of novel therapeutics, and thus deserve a mention on this review.
Author Response
Dear Reviewer,
Please find below a point by point reply to your comments.
We thank you for constructive suggestions and think that the manuscript has been considerably improved as a result, and hope that it will be now suitable for publication at the SciPharm.
On behalf of all authors,
Ekaterina Naumenko
Naumenko, Guryanov and Gomzikova present a comprehensive review of drug delivery systems. This subject is of high interest and this contribution may serve as a reference text. With that said I have some comments for the authors' consideration:
We thank you for positive assessment of our manuscript
Just below Table 1, there is a repeated sentence.
Answer: We fixed this point
Overall, the figures are appropriate but some of them have low resolution or illegible text. Plus, Figure 1 is cluttered with text; perhaps a redesign is due.
Answer: We redesigned the Figure 1
Major
While the presentation is very good and straightforward, I think there are some missing details that could improve the review. Such as further description of the procedures to generate the discussed nano systems.
Answer: We added requested information devoted to the methods of nano-sized vehicles fabrication.
Also, a missing aspect is a discussion or comparison between the presented methods. This would provide a wider context , while also providing perspectives on the directions in the field. On a similar note, there is little comment on the challenges or overarching difficulties associated with these platforms.
We added requested information throughout of all manuscript and in the new section 5. Comparative analysis of different drug delivery systems, their advantages and disadvantages
Finally, I noticed that the authors do not develop on the subject of dendrimers/polyplexes. Along with the mentioned platforms these also hold promise on the development of novel therapeutics, and thus deserve a mention on this review.
We added additional section devoted to the dendrimers using in drug delivery - 3. Dendrimers as anti-cancer drug delivery system

Round 2
Reviewer 2 Report
Comments and Suggestions for Authors
The authors have completed the reference list with some work but not with the most important one. They must use as keywords:
- drug loaded aptamer-conjugated liposomes; 5 - fluorouracil
- aptamer-functionalized polymeric nanocapsules; 5 - fluorouracil
- aptamer-functionalized nanocapsules; 5 - fluorouracil
Author Response
Dear Reviewer,
thank you for the consideration of our manuscript.
We added Reference suggested by you (Pharmaceutics 2021, 13, 866. https://doi.org/10.3390/pharmaceutics13060866). All additions and corrections of the text were highlighted by green.